# Superior Mechanical Properties of Invar36 Alloy Lattices Structures Manufactured by Laser Powder Bed Fusion

**DOI:** 10.3390/ma16124433

**Published:** 2023-06-16

**Authors:** Gongming He, Xiaoqiang Peng, Haotian Zhou, Guoliang Huang, Yanjun Xie, Yong He, Han Liu, Ke Huang

**Affiliations:** 1School of Materials Science & Engineering, Sichuan University, Chengdu 610065, China; hgming1988@163.com (G.H.); pengxiaoqiang@stu.scu.edu.cn (X.P.); zhouhaotian@stu.scu.edu.cn (H.Z.); hgl_lpbf_lab@stu.scu.edu.cn (G.H.); 2Sichuan Provincial Engineering Laboratory of Preparation Technology for Rare Earth & Vanadium-Titanium Based Functional Materials, Sichuan University, Chengdu 610065, China; 3PERA Global Co., Ltd., Beijing 100025, China; yanjun.xie@peraglobal.com; 4Science and Technology on Reactor Fuel and Materials Laboratory, Nuclear Power Institute of China, Chengdu 610213, China; hyscdx@163.com; 5SOlids inFormaTics AI-Laboratory (SOFT-AI-Lab), College of Polymer Science and Engineering, Sichuan University, Chengdu 610065, China; happylife@ucla.edu; 6Atlastech Additive Manufacuring Laboratory, School of Materials Science & Engineering, Sichuan University, Chengdu 610065, China

**Keywords:** Invar alloy, laser powder bed fusion (LPBF), triply periodic minimal surfaces (TPMS), energy absorption, anisotropy

## Abstract

Invar36 alloy is a low expansion alloy, and the triply periodic minimal surfaces (TPMS) structures have excellent lightweight, high energy absorption capacity and superior thermal and acoustic insulation properties. It is, however, difficult to manufacture by traditional processing methods. Laser powder bed fusion (LPBF) as a metal additive manufacturing technology, is extremely advantageous for forming complex lattice structures. In this study, five different TPMS cell structures, Gyroid (G), Diamond (D), Schwarz-P (P), Lidinoid (L), and Neovius (N) with Invar36 alloy as the material, were prepared using the LPBF process. The deformation behavior, mechanical properties, and energy absorption efficiency of these structures under different load directions were studied, and the effects and mechanisms of structure design, wall thickness, and load direction were further investigated. The results show that except for the P cell structure, which collapsed layer by layer, the other four TPMS cell structures all exhibited uniform plastic collapse. The G and D cell structures had excellent mechanical properties, and the energy absorption efficiency could reach more than 80%. In addition, it was found that the wall thickness could adjust the apparent density, relative platform stress, relative stiffness, energy absorption, energy absorption efficiency, and deformation behavior of the structure. Printed TPMS cell structures have better mechanical properties in the horizontal direction due to intrinsic printing process and structural design.

## 1. Introduction

Invar36 (64Fe, 36Ni, at.%) is a Fe-Ni alloy with a face-centered cubic lattice, renowned for its low expansion property below 250 °C (Curie temperature). The Invar36 alloy was initially discovered by French scientist Guillaume and earned him the Nobel Prize in Physics in 1920 [1,2]. To this day, Invar36 remains the most widely used low expansion alloy [3]. The most widely accepted theoretical explanation for its low expansion phenomenon is the magnetostriction effect. Below the Curie temperature, due to the appearance of ferromagnetism, the magnetic-induced lattice contraction counteracts part of the thermal expansion of the lattice as the temperature increases [4,5]. With its excellent low expansion, corrosion resistance, and superior mechanical properties, the Invar alloy is widely used in precision instruments, aerospace, electrical power, and other fields. With the continuous development in these fields, there is an increasing demand for high-performance components that meet lightweight requirements [6]. Therefore, it is particularly important to investigate new forming processes and structural designs to achieve high-performance components with the integration of structure and function.

A lattice structure is a class of lightweight structures with special mechanical, acoustic, thermal, and other physical properties constructed from one or more structural units (usually composed of rods, tubes, or plates) optimally combined in a specific way (periodic, topological, fractal, etc.) [7]. The performance of lattice structures is closely associated with cell characteristics, and their properties can be adjusted by manipulating the structural features of individual cells (e.g., cell connectivity or geometric dimensions). This allows for a wide range of design possibilities, making lattice structures an attractive choice for various applications, including aerospace, automotive, and biomedical industries [8,9].

TPMS is a kind of periodic smooth implicit surface with zero average curvature. Compared with other types of porous structures, TPMS has two significant advantages: (1) the overall TPMS porous structure can be precisely described using mathematical expressions, and basic properties, such as porosity and specific surface area, can be controlled directly using the function expression parameters; (2) the TPMS surface is extremely smooth without sharp turns or connection points of pillar-based lattice porous structures, and the overall structure is interconnected, which not only reduces stress concentration but also facilitates the removal of powder during the manufacturing process. Compared to solid alloys, TPMS structures have properties such as being lightweight and having high specific strength [10], and their mechanical properties can be adjusted in a wide range [7,11]. In addition, some unique lattice structures can also endow materials with specific non-mechanical properties, such as magnetic, electric, and optical properties [12]. Chen et al. [13] tested the compressive performance and shape memory effect of NiTi lattice structures of various sizes and gradients and found that the lattice structure can be controlled to achieve good energy absorption effects and shape memory effects. Yang et al. [14] conducted an analysis of the structure performance of NiTi alloy TPMS structures, and obtained the influence law of volume fraction and cell size on the compressive performance of NiTi alloy TPMS structures. Ravichander et al. [15] produced five types of TPMS structures: Gyroid, Diamond, Schwarz-P, Neovius, and Fisher-Koch S, using 316L materials, and analyzed their surface morphology, compressive performance, and energy absorption features, demonstrating that excellent stiffness and energy absorption capabilities can be obtained by hierarchical lattice structures. Zhang et al. [16] produced Schoen Gyroid surface structures of the TiB/Ti6Al4V system, finding that mechanical properties can be controlled by changing the ratio of TiB and Ti6Al4V. It can be seen that alloys designed into TPMS structures can not only achieve component lightweighting, but also modulate properties through structural design.

The traditional methods for preparing lattice structures mainly include investment casting, metal wire weaving, and wire cutting, but they have certain limitations: investment casting requires high fluidity of the metal liquid, and metal wire weaving and wire cutting require complex and delicate operations with low production efficiency and limited allowable structural complexity. There is, therefore, an urgent need to develop efficient, agile, and flexible manufacturing methods to prepare complex lattice structures [17].

Additive manufacturing, also known as 3D printing, is an intelligent manufacturing technology that uses digital models as the basis for adding materials layer by layer to create three-dimensional solid structures [18,19]. Laser powder bed fusion (LPBF), also known as selective laser melting (SLM), a typical metal additive manufacturing process, with characteristics of high efficiency, high resolution, waste reduction, and energy conservation, is particularly suitable for the preparation of fine and complex lattice structures [20,21]. LPBF technology was applied to microstructure regulation [21], dense gradient lattice structures [22], and TPMS lattice structures [23]. In addition, Invar36 alloy has good weldability and similar thermophysical properties between alloy components, which is particularly suitable for LPBF fabrication [24,25].

Due to the excellent damping performance of TPMS lattice structures [11,26,27] and the unique thermal expansion performance of Invar alloy, the combination of the two can yield multiple advantages, which can not only maintain the stress stability of Invar alloy structural components under loading conditions, but also reduce the thermal strain of TPMS lattice structures, ultimately obtaining a material with good structural stability under a wide range of temperatures and loads to meet specific application needs.

Currently, there are few studies on Invar alloy lattice structures. In this study, we conducted a systematic investigation of Invar TPMS structures and their mechanical properties, and explored their potential applications for the first time. Five different Invar alloy lattice structures with three wall thicknesses were prepared using LPBF technology, and compression tests were conducted in different directions. The printing parameter was optimized to achieve an exceptionally high quality, and the performance, including platform stress, stiffness, energy absorption, and energy absorption efficiency, was quantitatively calculated and systematically analyzed, and the effects and mechanism of structure, wall thickness, and load direction on the performance were discussed. Possible practical applications were also proposed.

## 2. Materials and Methods

### 2.1. Invar36 Alloy Powder

In this study, Invar36 alloy powder (ASTM grade 1, supplied by Hebei Jingye Additive Manufacturing Technology Co., Ltd., Shijiazhuang, China) was produced via gas atomization method. The particle size range is between 13 and 53 μm with a D50 value of 36.28 μm, as shown in Figure 1. The bulk density of the powder is 4.45 g/cm^3^, and the powder flowability measured by a Hall flow meter is good. A total of 50 g of powder can pass through a standard funnel with a diameter of 2.5 mm in only 15.24 s. It is observed that the majority of the gas-atomized Invar36 alloy powders exhibited a regular spherical shape, with a few smaller irregularly shaped particles. Moreover, most of the powder particles have no satellite powders on their surface, and the particle size distribution is in accordance with the normal distribution, satisfying the requirements of LPBF. Table 1 summarizes the chemical composition of Invar36 alloy powder, where the impurities such as C, O, P, and S content were all less than 0.01 wt.%. The powder exhibits a high degree of homogeneity, which is crucial in producing high-quality parts using LPBF.

### 2.2. Design of TPMS

A geometric model of TPMS was established using nTopology-4.1.3 software and STL files were exported for additive manufacturing. In this study, five types of TPMS cell structures with different unit structures were designed, namely Gyroid (G), Diamond (D), Schwarz-P (P), Lidinoid (L), and Neovius (N). Each structure was set with three wall thicknesses of 0.6 mm, 0.8 mm, and 1.0 mm, so that there were 15 different samples. The unit cell length of the designed cell structures in this study was 11 mm, and each lattice was composed of 2 × 2 × 2 unit cells. Therefore, the final model size was 22 mm × 22 mm × 22 mm, which satisfies the requirements of the ISO 13314:2011 standard [28] that the ratio of height to side length of rectangular specimens should be between 1 and 2. Figure 2 shows the designed models and solid samples fabricated by LPBF.

### 2.3. Preparation of TPMS

The TPMS structure is manufactured using an EOS-M290 machine (EOS, Krailling, Germany) and a 400 W single-mode fiber laser. Through a series of parameter optimization experiments, the molding process parameters of invar36 alloy powder were determined as follows: laser power of 150 W, scanning spacing of 70 μm, scanning speed of 800 mm/s, layer thickness of 30 μm, and laser beam diameter of 100 μm. Scan strategy is unidirectional scanning, with each layer rotating 67°. The process was carried out in an atmosphere filled with high-purity argon gas.

### 2.4. Testing and Characterization

The Archimedes method was used to determine the relative density of the samples, and quasi-static compression tests are conducted using an electronic universal testing machine (INSTRON 5985, Instron Corporation, Boston, MA, USA). The top plate was moved downward at a constant displacement rate of 0.05 mm/s, and the maximum strain was set to 75% according to the ISO 13314:2011 [28] standard. In order to investigate the effect of printing direction on structural performance, compression tests were carried out in both the building direction (Z direction) and the laser scanning direction (X direction). The strain was calculated by dividing the displacement of the top surface along the direction of the specimen’s loading by the original height of the specimen, and the compressive stress was obtained by dividing the load value by the apparent area of the crystal lattice structure. During the compression test, the deformation process of the specimen was recorded by video or photography.

Finally, the compressive yield strength, the plateau stress between 20% and 40% strain, and the unit volume energy absorption at 50% strain were calculated according to the ISO 13314:2011 standard. The yield strength was determined as the 0.2% offset yield strength. The plateau stress was calculated as the average stress between 20% and 40% compressive strain. The unit volume energy absorption at 50% strain was calculated as the area enclosed by the stress–strain curve and the *X*-axis at 50% strain.

## 3. Results

### 3.1. Analysis of Forming Quality

Since all the specimens were prepared under the same process parameters in the same batch, their quality parameters, such as density, microstructure, and roughness, should be almost the same. Figure 3 and Figure 4 show the metallographic image and the macro photograph of a structural pillar, respectively. From Figure 3a,b, it can be observed that there are almost no pores and cracks inside the pillar, with a density of 99.7% and a good continuity and metallurgical bonding of the scan track overlap. This indicates that the LPBF process parameters used in this study can produce cellular structures with high density. Furthermore, from Figure 4a–j, it can be seen that the cellular structure has a high forming quality and there are no obvious defects such as collapse and surface spheroidization at the support rod and the connection node.

The parameters of the prepared TPMS-cellular structure samples are shown in Table 2. The actual density of the printed sample is always greater than the theoretical density of the model, but the error is always below 10%. This is mainly caused by the adhesion of the un-melted powder particles to the surface of the sample during the printing process. For complex structures, in addition to the surface deviation caused by the enlargement of melt pool and the adhesion of powder particles, the cantilever structure is a key factor for local deviation [29]. Due to the lack of support, the thermal conductivity of the metal powder is low, and the melt pool will become larger and even sink into the underlying powder layer. In addition, the higher-temperature heat-affected zone will cause more partially melted powder to adhere to the layer below the working layer. As can be seen in Table 2 and Figure 5, the Gyroid-cellular structure has the highest accuracy retention rate, with the smallest average error between actual density and theoretical density, which is 4.56%. Under the same wall thickness conditions, the G and P-cellular structures have smaller apparent densities and great potential for weight reduction.

### 3.2. Mechanical Properties

#### 3.2.1. Compression Deformation Behavior

Figure 6 illustrates the compression deformation process of five different TPMS unit cell structures with a wall thickness of 0.6 mm. For the G, D, and L structures, their deformation processes are similar, with all layers collapsing simultaneously and undergoing uniform deformation in the 0–75% strain range. However, for the L unit cell structures with wall thicknesses of 0.8 mm and 1.0 mm, the volume fraction was too high, and the force reached the upper limit of the electron universal testing machine at 200 KN before the strain reached the predetermined value of 75%, so the experiment was stopped. For the P unit cell structures, regardless of the wall thickness, yielding occurred in a descending order, as the lower layers collapsed before compacting, and the upper layers collapsed after compacting. The possible reason for this is that the lower layer of the structure is subjected to its own gravitational force in addition to the load, making it reach the load carrying limit before the upper structure. For the N unit cell structures, when the wall thickness was 0.6 mm, their deformation behavior was similar to that of the P unit cell structures. When the wall thickness was increased to 0.8 mm and 1.0 mm, their deformation behavior was similar to that of the G, D, and L unit cell structures, with all layers collapsing simultaneously. This was attributed to the increase in wall thickness, the increase in the volume of load-bearing materials in the structure, and the uniform dispersion of the force, which enhances the stability of the structure. Finally, all samples did not fracture during the compression deformation process, and their deformation behavior in the Z and X directions was similar, indicating that the anisotropy of the structure had little effect on the macroscopic deformation process.

#### 3.2.2. Stress–Strain Curve

Figure 7 shows the stress–strain curves for five types of structures. From Figure 7a–e, it can be seen that stress–strain curves are similar for all types of specimens except for the P unit cell structure specimens. The stress–strain curve can be divided into three stages. The first stage is the elastic deformation stage, where the initial part of the curve is non-linear and slightly concave when a load is applied to the specimen at the beginning. This may be due to the roughness of the sample surface and size deviation, uneven loading on the unit cell and base plate, and the gap between equipment joints [30]. This phenomenon was also reported in other related literature [31,32]. After a small amount of strain, the unit cell structure enters the linear elastic region, and the stress–strain curve continues to rise to the stress peak. Then, the stress fluctuates slightly with strain, and the unit cell structure undergoes plastic collapse, forming a long yield plateau. The load acting on the specimen remains approximately constant until there is a noticeable increase, which is the second stage of the stress–strain curve plateau stress stage, also called the plastic collapse stage, and it is the main energy-absorbing stage of the material. Finally, under a larger compression strain, the entire structure completely collapses, and the prisms are pressed together to form a dense specimen, resulting in a sharp increase in stress, and the specimen enters the third stage of the stress–strain curve—the densification stage. For the P unit cell structure, the stress plateau stage of its stress–strain curve shows a changing trend of two increases and decreases, which corresponds to the two collapse processes of its compression deformation behavior.

The slope of the elastic section of the stress–strain curve, i.e., the structural stiffness, increases with an increase in wall thickness for the five types of TPMS structures (as shown in Figure 7). The maximum compressive strength can be reached within the range of 0–5% strain for each type of specimen, and the corresponding maximum strength increases with an increase in wall thickness. Meanwhile, with an increase in wall thickness, the yield plateau of the five structures keeps increasing. Samples of the same structure and wall thickness show inconsistency in the Z direction and X direction, where the slope of the elastic section in the X direction is always higher than that in the Z direction. In addition, the rising amplitude of the X direction is greater than that in the Z direction during the densification stage. Under the same wall thickness condition, the D unit cell structure has the longest yield plateau, while the L unit cell structure has the shortest yield plateau. For the N unit cell structure, samples with a wall thickness of 0.6 mm have a yield plateau similar to that of the P unit cell structure, with a certain degree of fluctuation (falling and then rising). With an increase in wall thickness, the yield plateau tends to be stable, similar to that of D and G structures.

#### 3.2.3. The Specific Plateau Stress and Specific Stiffness of Materials

Figure 8 shows a performance comparison of TPMS cell structure. Table 3 and Table 4 list the yield strength, elastic modulus, specific plateau stress, and specific stiffness values of the TPMS specimens. The yield strength is the average stress between 20% and 40% strain, which is an indicator of the strength of the porous material [28]. As the experiment used a fixed wall thickness, the density of the different specimens was different. To compare the performance of different structures, the specific plateau stress was defined as the ratio of the plateau strength to its apparent density, similar to the specific strength. The elastic modulus is the slope of the elastic portion of the stress–strain curve, and the specific stiffness is the ratio of the material’s elastic modulus to its apparent density.

As shown in Figure 8a, in the same structure, the plateau strength increases gradually with the increase in wall thickness. The L-cell structure with a wall thickness of 1.0 mm has the highest plateau strength of about 138.01 MPa, and the P-cell structure with a wall thickness of 0.6 mm has the lowest plateau strength of about 7.28 MPa. Similarly, as shown in Figure 8c, the specific plateau stress also increases gradually with the increase in wall thickness, but with a smaller magnitude, indicating that the main reason for the increase in strength is the increase in apparent density due to the increase in wall thickness. The specific plateau stress of the L-cell structure is lower than that of the N and D-cell structures, indicating that the D and N-cell structures have better load-bearing capacity under the same density. In addition, the specific plateau stress of the G-cell structure is also high and fluctuates slightly with the increase in wall thickness, while the specific plateau stress of the P-cell structure remains at a low level.

As shown in Figure 8b,d, the elastic modulus of the structure increases with the increase in wall thickness and the L-cell structure having a higher elastic modulus. However, among these five structures, the L-cell structure has the lowest specific stiffness. The 0.8 mm wall thickness specimen has a specific stiffness of about 0.65 GPa/(g·cm^−3^), and the D, N, and G-cell structures have higher specific stiffness values. There is a certain degree of anisotropy in the specific stiffness values of all cell structures in different directions, and the specific stiffness in the X direction is greater than that in the Z direction. The specific stiffness of the D and N-cell structures fluctuates significantly due to the influence of the structure’s wall thickness.

### 3.3. Energy Absorption Characteristics

The energy absorption capacity of the cellular structure is an important performance indicator for its applications. The energy absorption capacity of the cellular structure can be determined by numerically integrating the stress–strain curve under compression, which is denoted as the volumetric energy absorption (Wv) [33]:$$Wv=\int_{0}^{\varepsilon }\sigma \left(\varepsilon \right)d\varepsilon .$$

Specific energy absorption (*SEA*) is the energy absorbed per unit mass of the material [34]. *SEA* can be defined as:$$SEA=\frac{Wv}{m}.$$

In the equation, Wv represents the energy absorption in Joules, *σ* represents the compressive stress in MPa, and *ε* represents the compressive strain. The meaning of the equation is the area enclosed between the stress–strain curve and the *X*-axis (strain axis).

The energy absorption of TPMS-cellular structures at 50% strain is shown in Figure 9a and Table 5 and Table 6. At the same wall thickness, the P-cellular structure has the lowest energy absorption capacity, and the L-cellular structure has the highest energy absorption capacity, which is because of the fact that the energy absorption is related to the structure and density of the material. At the same wall thickness, the P-cellular structure has the lowest density, while the L-cellular structure has the highest density. For example, at a wall thickness of 1.0 mm, the densities of the P-cellular structure and L-cellular structure are 21.72% and 61.04%, respectively. At the same time, as shown in the stress–strain curve in Figure 7, the P-cellular structure has significant fluctuations in its stress–strain curve, resulting in lower energy absorption. Figure 9b shows the specific energy absorption of TPMS-cellular structures at 50% strain. It can be seen from the Figure 9 that the N and L-cellular structures have higher specific energy absorption, which increases with the increase in wall thickness. However, the specific energy absorption of the G, D, and P-cellular structures remains almost constant, indicating that the specific energy absorption is not significantly affected by the wall thickness.

In order to further illustrate the energy absorption capabilities of different structures, the energy absorption efficiency curves for each structure are shown in Figure 10. Energy absorption efficiency is defined by ISO 13314:2011 [28] as energy absorption divided by the product of the maximum compressive stress within the strain range and the size of the strain range, and it is used to describe the energy absorption capability of a material under compression. Based on the uniaxial stress–strain curve of crystalline materials, the energy absorption efficiency [33] is defined as:$$\eta \left(\varepsilon \right)=\frac{1}{\sigma_{m}\varepsilon }\int_{0}^{\varepsilon }\sigma \left(\varepsilon \right)d\varepsilon$$

*η* is the energy absorption efficiency of the material, *ε* is the strain at a specific moment during compression, and *σ_m_* is the stress corresponding to a specific *ε*. It can be seen that the energy absorption efficiency first decreases and then increases, and after a period of stability or fluctuation the curve shows an overall downward trend. Except for the P-cellular structure and the 0.6 mm wall thickness N-cellular structure, the energy absorption efficiency curves of all the structures are relatively stable, and the rapid decline in the later stage of the curve is mainly due to the rapid increase in the stress–strain curve during the densification stage. The energy absorption efficiency of all wall thicknesses of the D-cellular structure is mostly above 75% during the compression deformation process because the stress–strain curves of all wall thicknesses of the D-cellular structure have a longer plateau period. The energy absorption efficiency curve of the P-cellular structure is similar to that of the 0.6 mm wall thickness N-cellular structure, and the corresponding energy absorption efficiency curve fluctuates more due to the obvious shake in its stress–strain curve compared to the stress–strain curves of other structures. At the strain corresponding to the trough position of the stress–strain curve, the energy absorption efficiency of the P-cellular structure and the 0.6 mm N-cellular structure exceeds 1. At this point, the energy absorption of the lattice structure (i.e., the area enclosed by the stress–strain curve and the *x*-axis at this strain) may exceed the ideal energy absorption value (i.e., the area enclosed by the vertical line of the corresponding point on the stress–strain curve and the horizontal and vertical coordinate axes) as shown in Figure 10c,e. For the same structure, wall thickness and loading direction have no significant impact on the energy absorption efficiency of the structure.

## 4. Discussion

### 4.1. Effects of Lattice Structures

The mechanical properties of lattice structures depend on their topology, relative density, and base material. Compared to pillar-based lattice structures, TPMS structures have higher self-supporting capabilities. In addition, pillar-based lattice structures tend to have severe stress concentration at the pillar nodes. In contrast, TPMS surfaces have a more uniform stress distribution and higher stiffness [35]. Therefore, TPMS lattice structures have great advantages in structural lightweight applications.

The five TPMS structures designed in this study exhibit significantly different mechanical properties. As shown in Figure 11, the deformation mechanisms of G-cell, D-cell, and L-cell structures are dominated by bending mode (plateau stress is higher than yield stress), while the P-cell structure is dominated by stretching mode (plateau stress is lower than yield stress). At the same wall thickness, the P-cell structure has the lowest relative density, which is only 12.77% at a wall thickness of 0.6 mm. During deformation, stress concentration occurs around the supporting surface of the cavity, where deformation occurs first, as shown in Figure 6. The fluctuations in the platform period in Figure 11 are related to the deformation of the upper and lower layers of the structure. From Figure 8 and Figure 10, it can be seen that, compared with the other four structures, the P-cell structure has the lowest plateau stress and energy absorption capacity, and its energy absorption efficiency curve fluctuates greatly due to the unevenness of the deformation process. Therefore, the P-cell structure has the lowest structural stability.

At the same wall thickness, the L-cell structure has a higher plateau stress and stronger energy absorption capacity than the other structures, but its relative stiffness and resistance to deformation are weaker. Moreover, the energy absorption efficiency of the L-cell structure is not stable during the entire deformation process, and it drops sharply after a strain greater than 30%, which is mainly due to the relatively high relative density of the L-cell structure, short platform period, and too rapid densification process (as shown in Figure 11).

Finally, the D-cell, G-cell, and N-cell structures demonstrate excellent structural stability and good energy absorption performance. These three structures exhibit uniform deformation behavior in Figure 6 and all have a platform period greater than 50% strain in Figure 5. It is shown in Figure 9 and Figure 10 that these structures have high specific energy absorption and stable energy absorption efficiency. In terms of plateau stress and stiffness (Figure 8), the D-structure is more superior to the N-structure than the G-structure. It should be noted that the relative density and stability of the N-cell structure are reduced when the wall thickness decreases to 0.6 mm, and there is fluctuation in the deformation process (Figure 11a) and instability in the energy absorption efficiency (Figure 10e). Overall, the G-cell and D-cell structures have good comprehensive mechanical properties.

### 4.2. Effect of Wall Thickness

In the practical process of structural performance design, adjusting the wall thickness to control the structure’s apparent density and performance is an effective method. Therefore, it is necessary to evaluate the influence of structural wall thickness on performance. Since the wall thickness has a similar effect on the performance in the Z direction and X direction, the Z direction is taken as an example. The compressive yield strength was determined as 0.2% offset yield strength, and Table 7 shows the yield strength and plateau stress results of different wall thickness structures. For the G-cellular structure, the yield strength increased from 19.41 MPa to 38.29 MPa when the wall thickness was increased from 0.6 mm to 1.0 mm, an increase of about 97.27%. The numerical values of the D-cell, P-cell, L-cell, and N-cell structures were increased by about 84.65%, 195.52%, 106.26%, and 93.66%, respectively. The same phenomenon also occurred in the study of plateau stress, as the numerical values of the G-cell, D-cell, P-cell, L-cell, and N-cell structures were increased by 104.14%, 89.60%, 209.48%, 128.57%, and 128.87%, respectively, with the increase in wall thickness. The results indicate that increasing the wall thickness can significantly improve the yield strength and plateau stress of TPMS structures. This is because the increase in wall thickness results in an increase in the relative density of the structure, and the force can be distributed to more entities, which improves the stability of the structure. Similar results were obtained in Gyroid lattices by Zhang et al. [32].

From Figure 7, it can be seen that the deformation mechanisms of the G-cellular structure, D-cellular structure, and L-cellular structure are bending-dominated, and that of the P-cellular structure is stretching-dominated. The deformation mechanisms of these four structures are not affected by the change in wall thickness. However, as mentioned earlier, the deformation mechanism of the N-cellular structure changes with the wall thickness. When the wall thickness is 0.6 mm, the deformation mechanism of the N-cellular structure is stretching-dominated, while it becomes bending-dominated when the wall thickness increases to 0.8 mm or more. The possible reason is that when the wall thickness is small, the N-cellular structure is similar to the P-cellular structure, in which the stress concentration occurs on the supporting surface around the cavity, causing the supporting surface to yield. However, when the wall thickness is larger, the stability of the structure increases, and the stress concentration produced is not enough to cause a particular supporting surface to yield first. This is consistent with the results obtained by Zhou et al. [36], who designed a self-supporting lattice unit and gradually transformed the structure from bending-dominated to stretching-dominated by adjusting the connectivity of nodes in the unit. In essence, this is achieved by increasing support and improving the stability of the structure. Therefore, by changing the wall thickness, not only can the mechanical properties of the structure be adjusted, but also the deformation mechanism of the structure can be changed.

### 4.3. Anisotropy

As shown in Figure 7, the mechanical properties of the G-cell and N-cell structures in the X direction are significantly higher than those in the Z direction. Further analysis shows that in the samples with different wall thicknesses of the five TPMS structures, for the majority of the samples, the yield strength in the X direction is higher than that in the Z direction. For example, when the wall thickness is 0.6 mm, the yield strength of the G-cell structure increases by 10.15%, and the D, P, L, and N-cell structures increase by 2.37%, 48.75%, 0.97%, and 11.83%, respectively. This indicates that anisotropy may be common in TPMS structures. Weißmann et al. [37] and Choy [38] also, by rotating the cell structures at different angles and testing their mechanical properties in all directions, obtained similar results. The generation of anisotropy is related to the intrinsic laser processing characteristics and structural design of materials. First, during the LPBF preparation of TPMS structures, the growth direction and growth rate of grains in the TPMS structure are different in different directions due to the heat transfer conditions and laser scanning mode, resulting in different microstructural morphologies and mechanical properties anisotropy. Secondly, TPMS structures themselves may cause stress concentration in local positions during deformation and the difference in material properties in the X and Z directions would be amplified by the effect of stress concentration. The anisotropy of materials prepared by LPBF technology can be explained from the perspective of organizational characteristics [39], finite element analysis, or a combination of both [40,41]. Here, the anisotropy of performance is explained from the perspective of material microstructure and texture anisotropy.

EBSD analysis was performed on Invar36 alloy samples prepared by LPBF on the X-Y plane (X direction) and the X-Z plane (Z direction) to determine the grain morphology and orientation characteristics of the samples. As shown in Figure 12, the LPBF-formed Invar36 alloy has equiaxed grains in the X-Y direction, with an average grain size of approximately 30 μm, while in the X-Z direction, it has columnar grains with an average grain size of approximately 80 μm, with obvious anisotropy in the metallographic morphology. Further analysis indicates that this difference in microstructural morphology is determined by the metallurgical behavior of the molten pool in motion and the crystal nucleation/growth mechanism [42]. In fact, the shape of grains during the solidification process is determined by the instability of the solidification front and is related to the degree of undercooling. An increase in undercooling promotes the growth of equiaxed dendrites. The X-Y plane of LPBF-formed samples is a certain layer of processing surface, which will experience multiple re-melting and cooling during the printing process, as well as multiple thermal effects of subsequent printed layers. The X-Z plane of the sample is composed of cross-sections of multiple printed layers. The transverse and longitudinal cross-sectional forming methods are completely different, resulting in completely different microstructural morphologies. To illustrate the correspondence between texture anisotropy and performance anisotropy, a schematic diagram is shown in Figure 13. From Figure 13, it can be seen that when the cell is compressed along the *Z*-axis direction, the direction of force is approximately parallel to the columnar crystal direction on the XOZ plane during the compression process, and the grain boundary has weak resistance to deformation. This makes the material more prone to failure; therefore, the yield strength of the cell in the Z direction is lower. When the cell is compressed along the *X*-axis direction, the direction of force is perpendicular to the columnar crystals in the longitudinal cross-section, and the deformation process is hindered by equiaxed crystal grain boundaries distributed on the XOY plane. Compared with the *Z*-axis direction, the grain boundary has greater resistance to deformation, the deformation is more uniform, and the failure also requires a greater force. Therefore, the yield strength is higher when the cell is compressed in the *X*-axis direction.

In addition, there is a correlation between the anisotropy of performance and the internal stresses caused by non-equilibrium solidification [43]. Figure 14a,c show the KAM maps of the material, which are mainly used to characterize the stored energy and dislocation density and reflect the uniformity of deformation inside the material [44]. The blue indicates relatively uniform deformation, and the green indicates relatively concentrated stresses in the region. It can be seen that there are significant stress concentrations on both the X-Z and X-Y planes. In Figure 14b,d, the small-angle grain boundaries (LAGBs) are defined as those with an angle range of 2° to 10° and are represented by green lines, and the high-angle grain boundaries (HAGBs) are those with angles greater than 10° and are represented by black lines. The proportion of LAGBs on the X-Z plane is 61.9%, and it is 54.8% on the X-Y plane; both are higher than the proportion of HAGBs. It is evident from Figure 14b,d that the LAGBs are more likely to accumulate internal stresses, leading to significant lattice distortion between grains and making grain boundaries unstable and equivalent to failure-prone “defects”. Since there are fewer LAGBs on the X-Y plane than on the X-Z plane, the binding force between grain boundaries is more stable. Therefore, the yield strength of the sample is greater on the *X*-axis than that on the *Z*-axis.

### 4.4. Potential Applications of Different Structures

Due to the unique structural characteristics, TPMS unit cell structures have a wide range of applications in fields such as acoustics, optics, thermodynamics, and mechanics. However, when TPMS unit cell structures made of materials such as 316L and Ti6Al4V bear large loads, they tend to undergo layer collapse and shear failure, resulting in unstable energy absorption efficiency and other unstable behaviors [12,29]. In this study, TPMS unit cell structures are prepared using Invar36 alloy, which has high toughness and a low coefficient of thermal expansion. By coupling the excellent characteristics of lattice structures with the low coefficient of thermal expansion of Invar36, stable structures and damping functions were achieved under a wide range of temperatures and loads, greatly expanding its application range. Typical application cases include vibration damping brackets for high-precision optical instruments and lightweight precision molds.

Table 8 shows the performance characteristics and potential applications of the five TPMS unit cell structures. The D unit cell structure has the best overall performance. If considering the reduction in component weight and the constraint of minimum wall thickness in additive manufacturing, the G unit cell structure is more suitable, as it has 20% lower mass compared to the D unit cell structure for the same wall thickness and cell size. The P unit cell structure has non-uniform deformation characteristics, and due to its lower stress plateau and relative density, it can be used in lightweight structures and deformation warning components. The L unit cell structure has high relative density, high strength, and a short stress plateau period. The deformation mode of the N unit cell structure can be changed with wall thickness.

## 5. Conclusions

In this article, five types of Invar36 alloy TPMS cell structures with three different wall thicknesses were prepared using the LPBF process, and microstructural characterization and compression tests were performed to investigate the deformation behavior, mechanical properties, and energy absorption performance under different loading directions. We also explored the effects and mechanisms of structural design, wall thickness, and loading direction. The following conclusions can be drawn from this study:The TPMS cell structures made of Invar36 alloy with high toughness and low coefficient of thermal expansion, coupled with the excellent characteristics of lattice structures, achieve a large range of uniform deformation and outstanding damping and energy absorption properties;The five structural designs exhibit unique mechanical properties. The D-cell structure has the best overall performance. If considering structure lightweighting, the G-cell structure is more suitable because it has a 20% lower mass compared to the D-cell structure with the same wall thickness. The P-cell structure shows non-uniform deformation characteristics and a lower stress plateau and relative density. The L-cell structure has a high relative density, high strength, and a short stress plateau period. The failure mode of the N-cell structure can be altered with a change in wall thickness;The wall thickness can adjust the apparent density, stress plateau ratio, specific stiffness, energy absorption, energy absorption efficiency, and deformation mechanism of the structure. When the wall thickness is increased from 0.6 mm to 1.0 mm, the stiffness and stress plateau of the D, G, P, L, and N-cell structures increase by 74.37%, 42.33%, 97.03%, 69.84%, and 33.86%, respectively, and the stress plateau ratio increases by 104.14%, 89.60%, 209.48%, 128.57%, and 128.87%, respectively. The failure mode of the N-cell structure changes from the stretching dominant mode at 0.6 mm to the bending dominant mode at 0.8 mm and 1.0 mm;The performance of the TPMS structure in the horizontal direction is better than that in the vertical direction, which may be related to the anisotropy of the texture obtained after LPBF processing of the samples. The deformation of grains is more uniform and the yield strength is higher in the horizontal direction.

## Figures and Tables

**Figure 1 materials-16-04433-f001:**
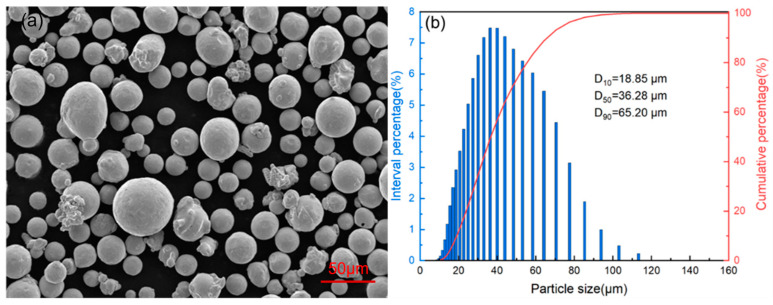
(**a**) Morphology of Invar36 powder, (**b**) particle size distribution.

**Figure 2 materials-16-04433-f002:**
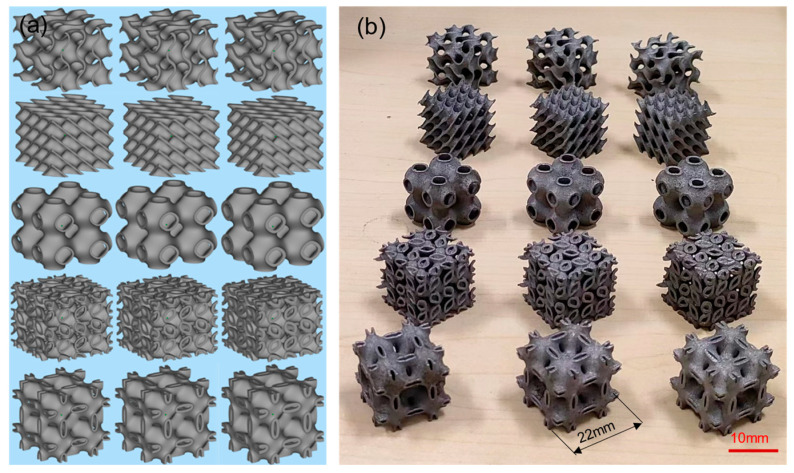
TPMS supports with different wall thicknesses (0.6 mm, 0.8 mm, and 1.0 mm). (**a**) CAD model image. (**b**) Actual sample image.

**Figure 3 materials-16-04433-f003:**
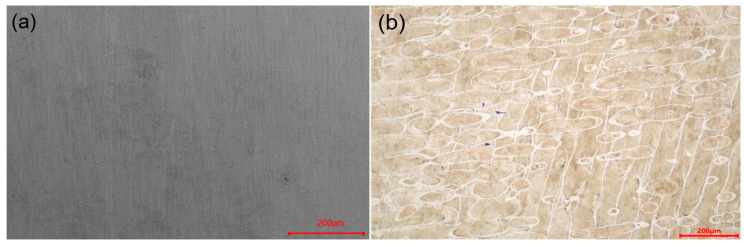
Metallographic image of the sample pillar and macro photographs of the TPMS structural sample in different directions. (**a**) Before etching, (**b**) after etching.

**Figure 4 materials-16-04433-f004:**
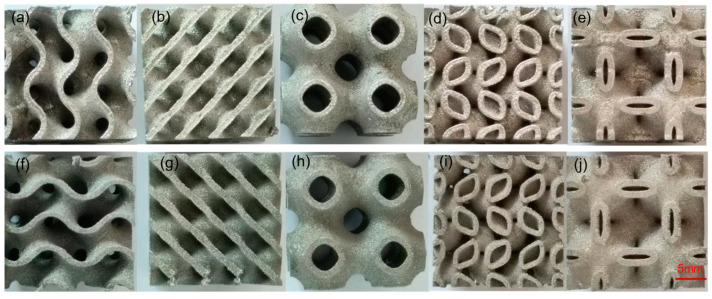
Macro photographs of the TPMS structural sample in different directions. (**a**–**e**) Photograph of the top surface of the sample, (**f**–**j**) photograph of the side surface of the sample.

**Figure 5 materials-16-04433-f005:**
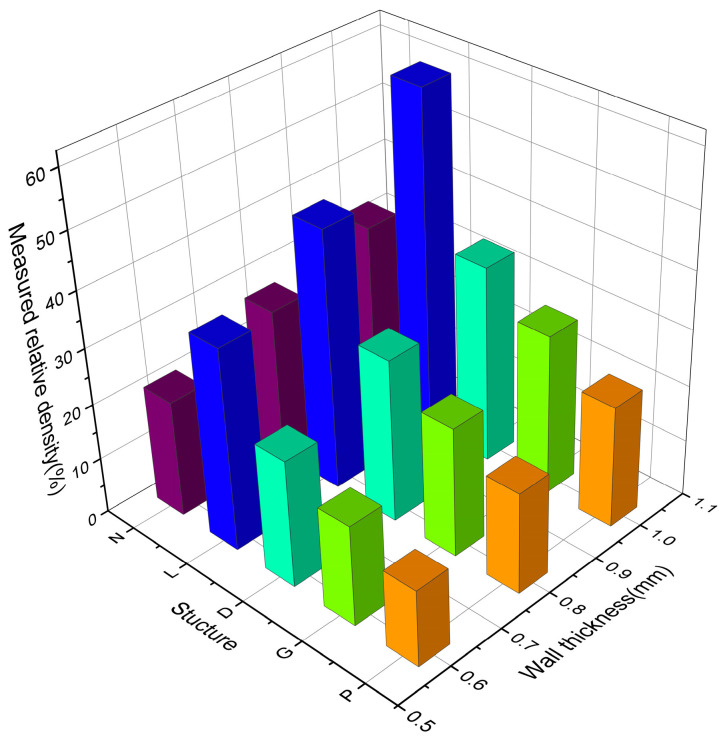
Relative density of TPMS samples with different wall thicknesses.

**Figure 6 materials-16-04433-f006:**
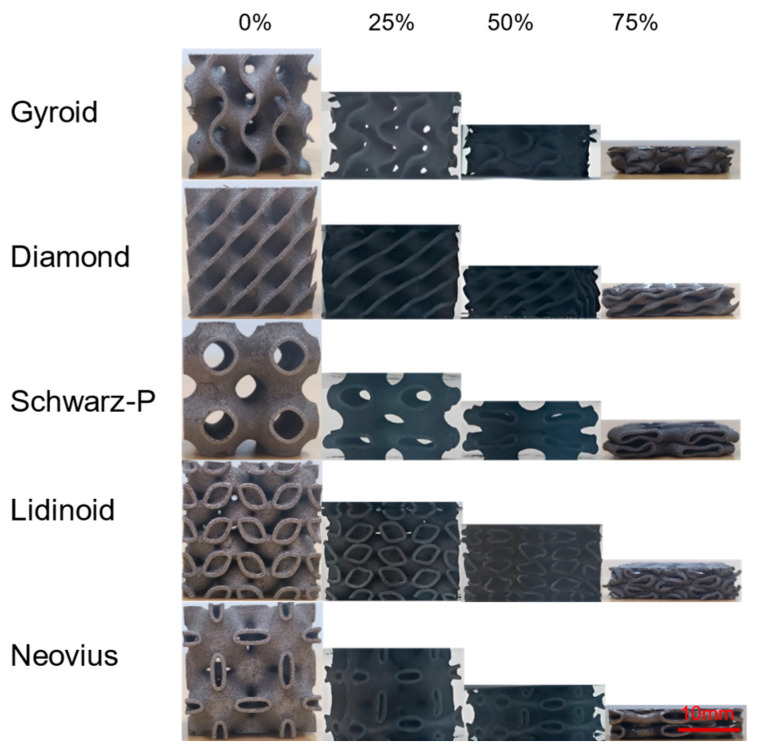
The compression process of TPMS unit cell structures with a wall thickness of 0.6 mm.

**Figure 7 materials-16-04433-f007:**
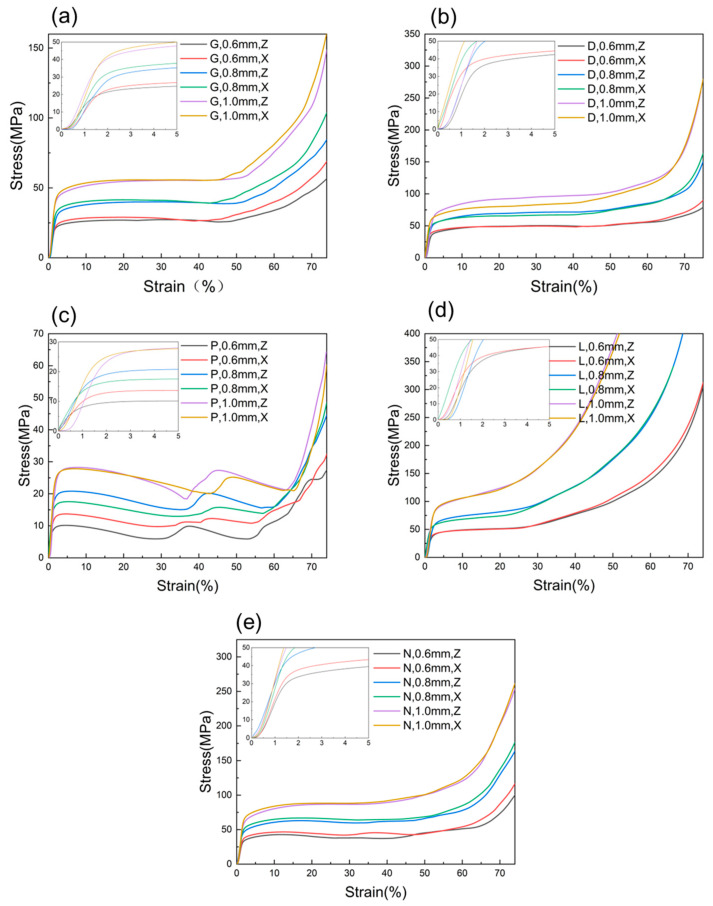
Stress–strain curves for specimens with different wall thicknesses. (**a**) Gyroid, (**b**) Diamond, (**c**) Schwarz-P, (**d**) Lidinoid, and (**e**) Neovius.

**Figure 8 materials-16-04433-f008:**
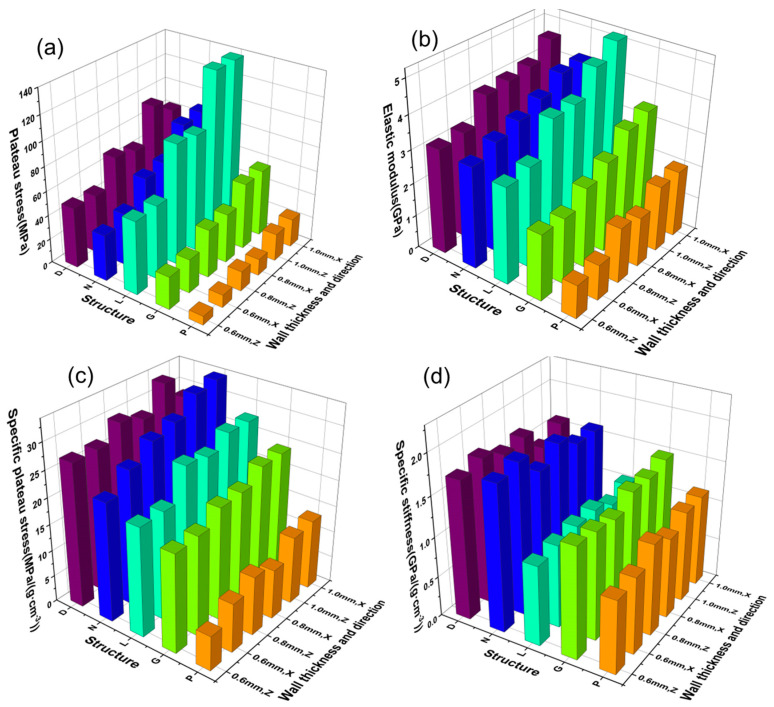
Comparison of properties of TPMS specimens with different wall thicknesses. (**a**) Plateau stress of TPMS specimens. (**b**) Elastic modulus of TPMS specimens. (**c**) Specific plateau stress of TPMS specimens. (**d**) Specific stiffness of TPMS specimens.

**Figure 9 materials-16-04433-f009:**
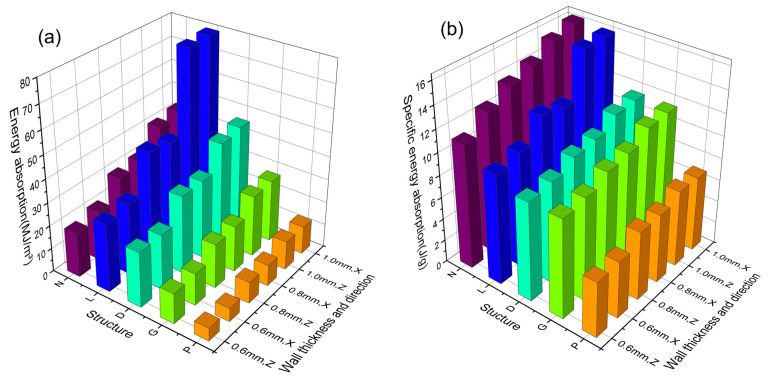
The energy absorption and specific energy absorption of TPMS specimens at 50% strain for different wall thicknesses. (**a**) The energy absorption of TPMS specimens. (**b**) The specific energy absorption of TPMS specimens.

**Figure 10 materials-16-04433-f010:**
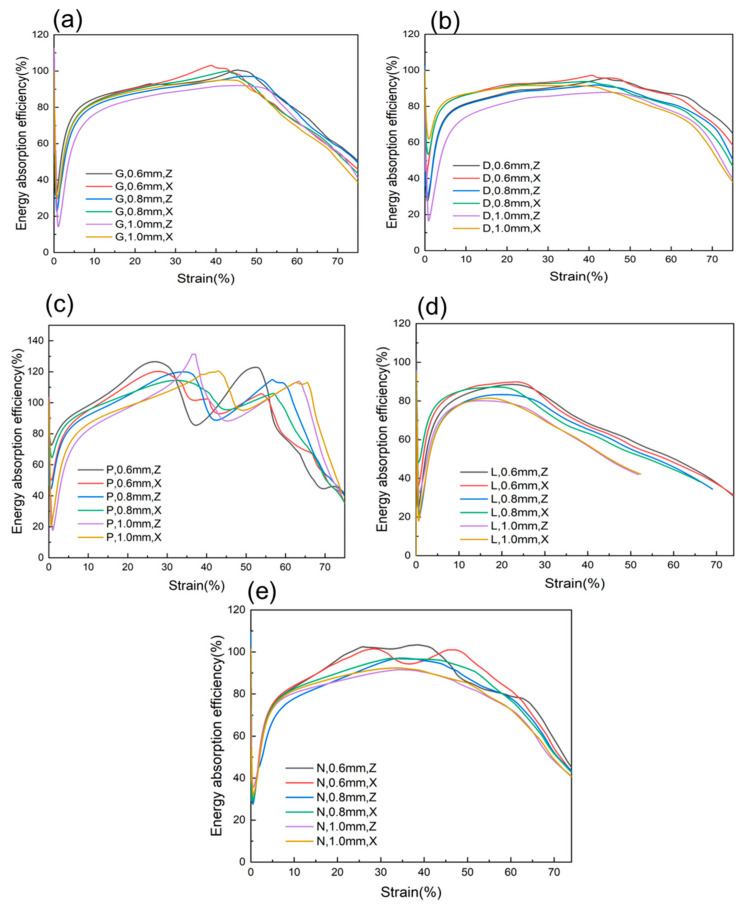
Energy absorption efficiency curves of samples with different wall thicknesses. (**a**) Gyroid structure. (**b**) Diamond structure. (**c**) Schwarz-P structure. (**d**) Lidinoid structure. (**e**) Neovius structure.

**Figure 11 materials-16-04433-f011:**
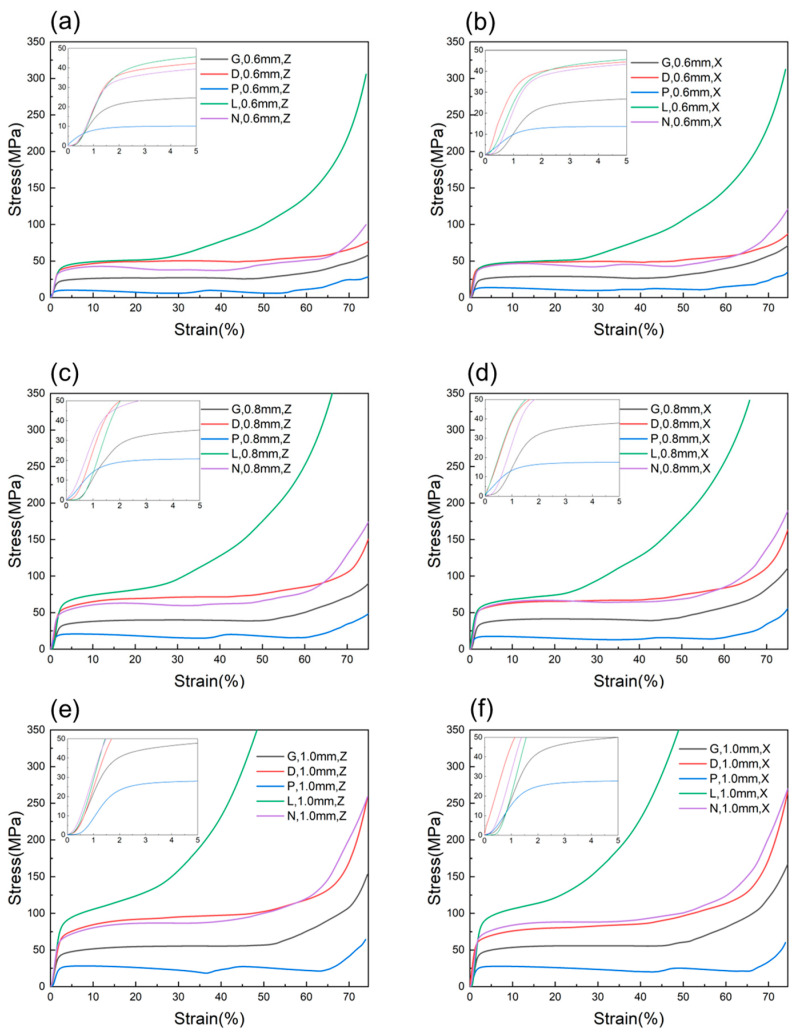
Stress–strain curves of samples with the same wall thickness. (**a**) Stress–strain curve in the Z direction of a sample with 0.6 mm wall thickness. (**b**) Stress–strain curve in the X direction of a sample with 0.6 mm wall thickness. (**c**) Stress–strain curve in the Z direction of a sample with 0.8 mm wall thickness. (**d**) Stress–strain curve in the X direction of a sample with 0.8 mm wall thickness. (**e**) Stress–strain curve in the Z direction of a sample with 1.0 mm wall thickness. (**f**) Stress–strain curve in the X direction of a sample with 1.0 mm wall thickness.

**Figure 12 materials-16-04433-f012:**
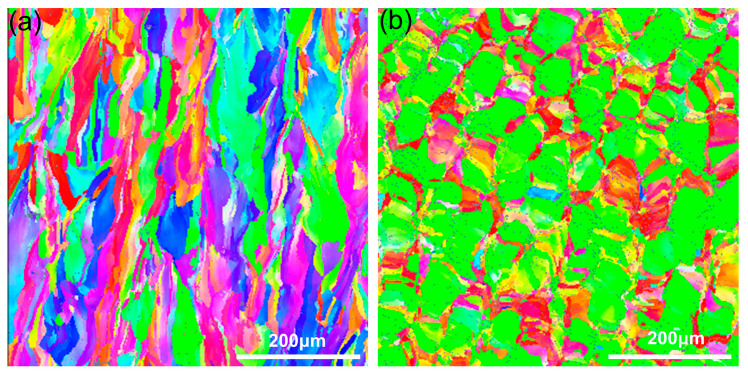
Grain morphology of Invar36 alloy samples in different directions: (**a**) X-Z plane (Z direction), (**b**) X-Y plane (X direction).

**Figure 13 materials-16-04433-f013:**
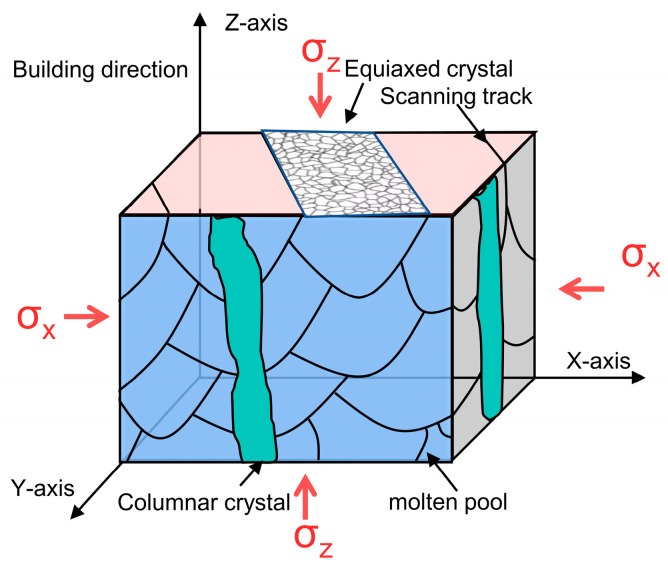
The relationship between the compression direction and the position of the melt pool structure during compression of the formed Invar36 alloy.

**Figure 14 materials-16-04433-f014:**
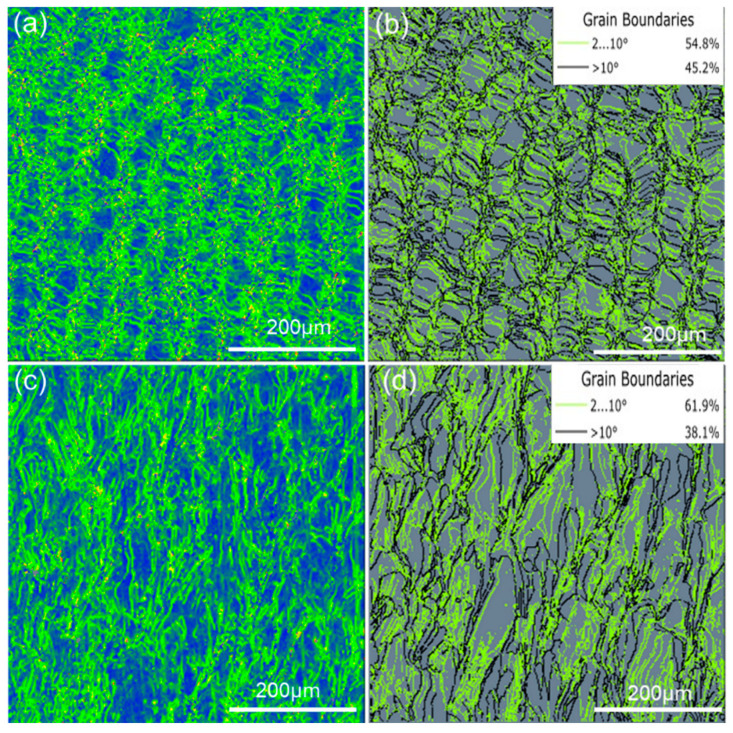
The KAM maps (**a**,**c**) and grain boundary maps (**c**,**d**) of the as-formed Invar36 alloy: (**a**,**b**) X-Y plane, (**c**,**d**) X-Z plane.

**Table 1 materials-16-04433-t001:** Chemical composition of Invar36 alloy spherical powder.

Element	C	Si	Mn	P	Ni	S	Cr	Co	O
Wt.%	0.011	0.13	0.29	0.004	35.8	0.002	0.005	0.03	0.000284

**Table 2 materials-16-04433-t002:** Weight and density of the designed sample.

Type of Sample	Wall Thickness (mm)	Sample Mass (g)	Measured Relative Density (%)	Designed Relative Density (%)	Error (%)
Gyroid	0.6	15.43	17.89	16.75	6.81
	0.8	19.94	23.11	22.28	3.74
	1.0	24.66	28.59	27.72	3.14
Diamond	0.6	19.55	22.66	20.75	9.21
	0.8	25.15	29.16	27.54	5.87
	1.0	30.46	35.32	34.22	3.22
Schwarz-P	0.6	11.79	13.67	12.77	7.01
	0.8	15.58	18.07	16.99	6.35
	1.0	18.74	21.72	21.19	2.52
Lidinoid	0.6	31.13	36.10	33.39	8.11
	0.8	39.91	46.27	43.95	5.28
	1.0	52.64	61.04	55.66	9.66
Neovius	0.6	18.00	20.87	19.05	9.58
	0.8	23.20	26.90	25.28	6.40
	1.0	28.14	32.62	31.37	3.99

**Table 3 materials-16-04433-t003:** Plateau stress, specific plateau stress, elastic modulus, and specific stiffness values in the Z direction for TPMS specimens.

	G			D			P			L			N		
	0.6 mm	0.8 mm	1.0 mm	0.6 mm	0.8 mm	1.0 mm	0.6 mm	0.8 mm	1.0 mm	0.6 mm	0.8 mm	1.0 mm	0.6 mm	0.8 mm	1.0 mm
Plateau stress (MPa)	27.04	39.84	55.20	50.00	70.72	94.80	7.28	16.3	22.53	60.38	99.19	138.01	38.03	62.40	87.04
Specific plateau stress (MPa/(g·cm^−3^))	18.66	21.29	23.84	27.24	29.94	33.14	6.58	11.14	12.80	20.65	26.47	27.92	22.49	28.64	32.94
Elastic Modulus (GPa)	2.01	2.42	3.29	3.17	3.98	4.14	1.01	1.67	1.99	2.92	4.02	4.77	3.07	3.50	4.27
Specific stiffness (GPa/(g·cm^−3^))	1.39	1.29	1.42	1.72	1.69	1.45	0.92	1.14	1.13	1.00	1.07	0.96	1.82	1.61	1.61

**Table 4 materials-16-04433-t004:** Plateau stress, specific plateau stress, elastic modulus, and specific stiffness values in the X direction for TPMS specimens.

	G			D			P			L			N		
	0.6 mm	0.8 mm	1.0 mm	0.6 mm	0.8 mm	1.0 mm	0.6 mm	0.8 mm	1.0 mm	0.6 mm	0.8 mm	1.0 mm	0.6 mm	0.8 mm	1.0 mm
Plateau stress (MPa)	27.96	40.87	55.2	49.17	66.44	82.74	10.41	13.65	23.38.	61.03	96.19	137.96	43.76	64.95	88.82
Specific plateau stress (MPa/(g·cm^−3^))	18.92	21.48	23.90	27.28	28.51	28.60	9.28	9.43	13.25	20.77	25.70	27.81	25.83	29.87	33.64
Elastic Modulus (GPa)	1.99	2.74	3.47	3.26	4.05	4.64	1.01	1.67	1.99	3.05	4.05	5.18	3.19	3.86	4.27
Specific stiffness (GPa/(g·cm^−3^))	1.35	1.44	1.49	1.81	1.74	1.60	0.96	1.04	1.14	1.04	1.08	1.04	1.89	1.78	1.62

**Table 5 materials-16-04433-t005:** The energy absorption and specific energy absorption in the Z direction of TPMS specimens at 50% strain.

	G			D			P			L			N		
	0.6 mm	0.8 mm	1.0 mm	0.6 mm	0.8 mm	1.0 mm	0.6 mm	0.8 mm	1.0 mm	0.6 mm	0.8 mm	1.0 mm	0.6 mm	0.8 mm	1.0 mm
Wv (MJ/m^3^)	12.91	18.84	26.23	23.60	33.50	43.89	5.52	9.04	12.18	28.91	46.55	78.80	18.94	30.17	41.26
*SEA* (J/g)	8.91	10.06	11.33	12.86	14.18	15.34	4.99	6.18	6.92	9.89	12.42	15.94	11.20	13.85	15.61

**Table 6 materials-16-04433-t006:** The energy absorption and specific energy absorption in the X direction of TPMS specimens at 50% strain.

	G			D			P			L			N		
	0.6 mm	0.8 mm	1.0 mm	0.6 mm	0.8 mm	1.0 mm	0.6 mm	0.8 mm	1.0 mm	0.6 mm	0.8 mm	1.0 mm	0.6 mm	0.8 mm	1.0 mm
Wv (MJ/m^3^)	13.64	19.67	26.70	23.74	33.67	45.40	5.74	8.99	11.99	30.84	44.79	79.31	21.59	31.59	42.89
*SEA* (J/g)	9.23	10.34	11.46	13.17	14.45	15.70	5.12	6.21	6.79	10.50	11.96	15.99	12.75	14.53	16.24

**Table 7 materials-16-04433-t007:** Yield strength and plateau stress of TPMS samples.

		G			D			P			L			N		
		0.6 mm	0.8 mm	1.0 mm	0.6 mm	0.8 mm	1.0 mm	0.6 mm	0.8 mm	1.0 mm	0.6 mm	0.8 mm	1.0 mm	0.6 mm	0.8 mm	1.0 mm
Z direction	yield strength (MPa)	19.41	25.28	38.29	35.84	50.31	66.18	7.59	16.36	22.43	33.89	51.59	69.90	31.54	43.20	61.08
	Plateau stress (MPa)	27.04	39.84	55.20	50.00	70.72	94.80	7.28	16.3	22.53	60.38	99.19	138.01	38.03	62.40	87.04
X direction	yield strength (MPa)	21.38	31.84	41.23	36.69	48.82	58.98	11.29	13.99	22.56	34.22	45.87	74.71	35.27	49.33	64.61
	Plateau stress (MPa)	27.96	40.87	55.69	49.17	66.44	82.74	10.41	13.65	23.38	61.03	96.23	137.96	43.76	64.95	88.82

**Table 8 materials-16-04433-t008:** Performance and potential applications of TPMS structures.

Type of Sample	Performance	Potential Applications
Gyroid	Relative low density, high ductility, high strength, excellent energy absorption capacity, and low expansion coefficient.	Lightweight structures, anti-collision energy absorption devices, precision instruments, vibration damping, and noise reduction.
Diamond	High ductility, high strength, excellent energy absorption capacity, and low expansion coefficient.	Lightweight structures, crash energy absorption devices, precision instruments, and vibration and noise reduction.
Schwarz-P	High ductility, low strength, relative low density, non-uniform deformation, and low coefficient of thermal expansion.	Lightweight structures, deformation warning components, crash energy absorption devices, and vibration and noise reduction.
Lidinoid	High ductility, high strength, high relative density, short stress plateau period, and low coefficient of thermal expansion.	Lightweight structures, crash energy absorption devices, and vibration and noise reduction.
Neovius	High ductility, excellent energy absorption capability, convertible failure mode, and low coefficient of thermal expansion.	Lightweight structures, crash energy absorption devices, precise instruments, and vibration and noise reduction.

## Data Availability

Not applicable.

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
