# Peer review of "Superior Mechanical Properties of Invar36 Alloy Lattices Structures Manufactured by Laser Powder Bed Fusion"

_materials, 2023, doi:10.3390/ma16124433_

Round 1

Reviewer 1 Report

The quality of english and presentation throughout are very high. The content is largely geared towards mechanical engineering and metallurgy which, as stated in the comments of my acceptance, are somewhat outside my area of expertise so I will confine my comments to the ares in which I have extensive experience, those of LPBF and design of lattice structures. With regards to these, the content overall is quite good but I would have some minor comments on areas that could be improved. These points are as follows:

  1. The powder characterisation is a bit short for such an important component of the experimental results. Ideally the powder characterisation would include rheology measurements, in particular a flowability measurement would be advisable here given the importance of powder flowability to the mechanical properties of the resulting parts produced (DOI: 10.1016/j.addma.2019.100929). In addition, the number of defects in the final part is known to be heavily influenced by the flowability and apparent density of the feedstock powder (DOI: 10.3390/ma11122386). Of course, particle size and distribution influence both of these factors but direct measurements of these powder characteristics are important to ensure the reproducibility of the mechanical results presented.

  2. The scan strategy is insufficiently described and what description is present possibly contains a typo? In section 2.3 the author states they have a scan spacing of 70mm, which would be larger than the dimensions listed for the entire part. I suspect this was supposed to be 70 microns. Also, no actual scanning strategy is cescribed beyond this. Was the scan unidirectional, bidirectional, checkerboard, helical, spiral, contour, or some other less common strategy? This information should be included in the description of the print parameters.

  3. Table 2 shows surprisingly large discrepancies between the expected relative density and the measured relative density. Even more perplexingly, it is consistently above the expected rather than below. The authors offer a plausible potential explanation but have not ruled out another possible explanation that, personally, I have encountered when printing (albeit smaller) topological structures: inaccuracy resulting from CAD model tesselation. The authors state that they exported from ntopology to an stl file, which presumably means they then sliced that stl file in separate software such as netfabb. If this is correct, then they havent reported the dimensional accuracy of the tesselated stl models exported. In my experience, many CAD softwares default to quite large triangles and relatively low accuracy for the kinds of applications we develop for, which can result in high dimensional inaccuracy in the stl models produced. For this reason, i would recommend using a point-normal tesselated format in future (e.g: the .amf file format) or, for even better accuracy, slicing directly in nTopology for maximum shape accuracy. As it stands, the authors should at least examine the stl files produced to rule this dimensional inaccuracy out as a potential source of the excess mass.

Author Response

Thank you for your encouraging letter and valuable comments concerning our manuscript entitled “Superior mechanical properties of Invar 36 alloy lattices structures manufactured by Laser powder bed fusion (ID:2399649)”. Those comments are all very helpful for revising and improving our paper, as well as guiding our future research. We have studied comments carefully and have made corrections and clarifications per reviewer’s suggestions and recommendations. Revised portion are marked in red in the paper. The main corrections in the paper and the responds to the reviewer's comments are as flowing:

Responds to the reviewer's comments:

Reviewer #1:

Q1: The powder characterization is a bit short for such an important component of the experimental results. Ideally the powder characterization would include rheology measurements, in particular a flowability measurement would be advisable here given the importance of powder flowability to the mechanical properties of the resulting parts produced (DOI: 10.1016/j.addma.2019.100929). In addition, the number of defects in the final part is known to be heavily influenced by the flowability and apparent density of the feedstock powder (DOI: 10.3390/ma11122386). Of course, particle size and distribution influence both of these factors but direct measurements of these powder characteristics are important to ensure the reproducibility of the mechanical results presented.

A1:Thank you very much for your comments. Powder properties as critical influencing factors really need to be more fully characterized, so I have added the following on page 3, paragraph 3 of the article: The bulk density of the powder is 4.45 g/cm3, and the powder flowability measured by a Hall flow meter is good. 50g of powder can pass through a standard funnel with a diameter of 2.5mm in only 15.24 seconds.

Q2:The scan strategy is insufficiently described and what description is present possibly contains a typo? In section 2.3 the author states they have a scan spacing of 70mm, which would be larger than the dimensions listed for the entire part. I suspect this was supposed to be 70 microns. Also, no actual scanning strategy is cescribed beyond this. Was the scan unidirectional, bidirectional, checkerboard, helical, spiral, contour, or some other less common strategy? This information should be included in the description of the print parameters.

A2:A scan spacing of 70mm is indeed a typo, the correct spacing should be 70μm The scanning strategy is unidirectional with 67° rotation between layers. The above has been supplemented and modified in the second paragraph on page 4 of the article.

Q3: Table 2 shows surprisingly large discrepancies between the expected relative density and the measured relative density. Even more perplexingly, it is consistently above the expected rather than below. The authors offer a plausible potential explanation but have not ruled out another possible explanation that, personally, I have encountered when printing (albeit smaller) topological structures: inaccuracy resulting from CAD model tesselation. The authors state that they exported from ntopology to an stl file, which presumably means they then sliced that stl file in separate software such as netfabb. If this is correct, then they havent reported the dimensional accuracy of the tesselated stl models exported. In my experience, many CAD softwares default to quite large triangles and relatively low accuracy for the kinds of applications we develop for, which can result in high dimensional inaccuracy in the stl models produced. For this reason, i would recommend using a point-normal tesselated format in future (e.g: the .amf file format) or, for even better accuracy, slicing directly in nTopology for maximum shape accuracy. As it stands, the authors should at least examine the stl files produced to rule this dimensional inaccuracy out as a potential source of the excess mass.

A3: Thank you very much for your comments, which are very insightful. The following is a description of the STL model used in this experiment: the mesh size of 0.6mm, 0.8mm and 1.0mm wall thickness is 0.2mm, 0.25mm and 0.3mm, respectively. The accuracy should be high enough.

Thanks again for your encouraging and valuable comments.

Reviewer 2 Report

General comments:

The paper investigates the compression properties of the lattice structures printed using metal via selective laser melting.

Specific comments:

1. Figures 8 and 9, although 3d plots allow the readers to see the general trends, they make retrieving the specific data point difficult. it also makes the texts on the axes hard to read.

2. font size of the axes title is too small, suggest enlarging it.

3. for images with specimens, suggest adding a scale bar.

4. i believe it is using selective laser melting technique, suggest using this term (SLM) instead of LPBF. as it is the term used in the ASTM.

Author Response

Thank you for your encouraging letter and valuable comments concerning our manuscript entitled “Superior mechanical properties of Invar 36 alloy lattices structures manufactured by Laser powder bed fusion (ID:2399649)”. Those comments are all very helpful for revising and improving our paper, as well as guiding our future research. We have studied comments carefully and have made corrections and clarifications per reviewer’s suggestions and recommendations. Revised portion are marked in red in the paper. The main corrections in the paper and the responds to the reviewer's comments are as flowing:

Responds to the reviewer's comments:

Reviewer #2:

Q1:Figures 8 and 9, although 3d plots allow the readers to see the general trends, they make retrieving the specific data point difficult. it also makes the texts on the axes hard to read.

A1:Thank you very much for raising this question. Due to the large amount of data in our experiment, choosing other types of graphs may not present the trend well. At the same time, considering the difficulty of data point retrieval, we have added table 3 and 4 with all data points above both Figure 8 and Figure 9 to facilitate readers' reference. Reading text on the axes does require higher reading skills for 3D graphs compared to 2D graphs, however, we already tried very hard to show axes texts clearly while keeping all points in the figures visible.

Q2: font size of the axes title is too small, suggest enlarging it.

A2:Thank you for your valuable suggestion. The axis titles have been modified to make them more prominent(Figure 7, Figure 8, Figure 10, Figure 11).

Q3: for images with specimens, suggest adding a scale bar.

A3: Thank you very much for your comments, and I have added a scale bar on Figure 2, Figure 4, Figure 6.

Q4:I believe it is using selective laser melting technique, suggest using this term (SLM) instead of LPBF. as it is the term used in the ASTM.

A4:Thank you very much for your question. It is true that terminology should be consistent with the standard in article writing. SLM was used widely before. However, refer to ISO/ASTM 52900: Additive manufacturing — General principles — Fundamentals and Vocabulary. In this standard, it uses Laser powder bed fusion (LPBF) currently.

Thanks again for your encouraging and valuable comments.

Reviewer 3 Report

The submitted paper shows a detailed investigation of the mechanical properties of the Invar 36 alloy lattice structure produced by the LPBF technique. The manuscript is well-written and brings interesting results. I really enjoyed reading this work due to its scientific sound and merit. In order to contribute to the improvement of the present research, some minor review is required. My raised points can be seen as follow:

Materials and methods:

Metallography images are shown, how was the procedure for that? What was the solution used to etch the samples?

2.2 section

"laser power of 150W, scanning spacing of 70mm, scanning speed of 800mm/s," for the scanning spacing, the unit should micrometers instead fo millimeters.

Section 3.2.1

"However, for the L unit cell structures with wall thicknesses of 0.8mm and 1.0mm, the volume fraction was too high, and the force reached the upper limit of the electron universal testing machine at 200KN before the strain reached the predetermined value of 75%, so the experiment was stopped."

Authors must arrange a way like supplementary data to show the figures regarding the samples with the wall thickness of 0.8 and 1.0 mm

Author Response

Thank you for your encouraging letter and valuable comments concerning our manuscript entitled “Superior mechanical properties of Invar 36 alloy lattices structures manufactured by Laser powder bed fusion (ID:2399649)”. Those comments are all very helpful for revising and improving our paper, as well as guiding our future research. We have studied comments carefully and have made corrections and clarifications per reviewer’s suggestions and recommendations. Revised portion are marked in red in the paper. The main corrections in the paper and the responds to the reviewer's comments are as flowing:

Responds to the reviewer's comments:

Reviewer #3:

Q1: M`aterials and methods: Metallography images are shown, how was the procedure for that? What was the solution used to etch the samples?

A1: Thank you for your question. The metallographic processing procedure is as follows: firstly, rough grinding the sample with sandpapers in order of 200, 400, 600, 800, 1000, 1200, 1500, 2000, and 3000 grit; then polishing with diamond solution; cleaning with alcohol and drying; finally, etching with the etchant solution for 3-4 minutes, then taking out the sample, cleaning with alcohol, and drying, and observing the metallographic structure. The etchant solution is prepared as follows: 3g CuCl2+50ml HCl+50ml C2H5OH+50ml H2O.

Q2: 2.2 section "laser power of 150W, scanning spacing of 70mm, scanning speed of 800mm/s," for the scanning spacing, the unit should micrometers instead of millimeters.

A2: Thank you very much for pointing out this issue. There was indeed a writing error here. The correct scanning interval is 70μm. The content has been modified in the second paragraph of page 4 of the article.

Q3:Section 3.2.1 "However, for the L unit cell structures with wall thicknesses of 0.8mm and 1.0mm, the volume fraction was too high, and the force reached the upper limit of the electron universal testing machine at 200KN before the strain reached the predetermined value of 75%, so the experiment was stopped."

Authors must arrange a way like supplementary data to show the figures regarding the samples with the wall thickness of 0.8 and 1.0 mm.

A3:Thank you for raising this question. We will add compression deformation diagrams of L type Lattice structures with wall thicknesses of 0.8mm and 1.0mm. Please see the deformation images and stress-strain curves of L type lattice (same to Fig. 7 (d)) in attached file.

The compression performance was tested by Inston5859 electronic universal testing machine with the maximum load force of 200kN. During the compression testing of L-structure with wall thickness of 1.0 mm, when deformation is 50%, the compression force was 200kN and reached the limit of the equipment, and the experiment was terminated. For thickness 0.8mm, the force limit 200kN was reached at deformation 65%.

Thanks again for your encouraging and valuable comments.
